# The Complete Chloroplast Genomes of *Blepharoglossum elegans* and *B. grossum* and Comparative Analysis with Related Species (Orchidaceae, Malaxideae)

**DOI:** 10.3390/genes14051069

**Published:** 2023-05-11

**Authors:** Wenting Yang, Kunlin Wu, Lin Fang, Songjun Zeng, Lin Li

**Affiliations:** 1Key Laboratory of South China Agricultural Plant Molecular Analysis and Genetic Improvement, South China Botanical Garden, Chinese Academy of Sciences, Guangzhou 510650, China; yangwenting@scbg.ac.cn (W.Y.); zengsongjun@scib.ac.cn (S.Z.); 2Guangdong Provincial Key Laboratory of Applied Botany, South China Botanical Garden, Chinese Academy of Sciences, Guangzhou 510650, China; 3University of Chinese Academy of Sciences, Beijing 100049, China

**Keywords:** *Blepharoglossum*, chloroplast genome, comparative genomic, Orchidaceae, phylogeny

## Abstract

*Blepharoglossum* is a rare orchid genus of the Malaxidinae primarily distributed in tropical Pacific islands, with several species occurring in the Taiwan and Hainan Islands of China. Currently, the monophyletic status of *Blepharoglossum* has been challenged, and the phylogenetic relationships among its allied groups have remained unresolved with traditional DNA markers. In this study, we initially sequenced and annotated the chloroplast (cp) genomes of two *Blepharoglossum* species, *Blepharoglossum elegans* (Lindl.) L. Li and *Blepharoglossum grossum* (Rchb.f.) L. Li. These cp genomes of *Blepharoglossum* share the typical quadripartite and circular structure. Each of the genomes encodes a total of 133 functional genes, including 87 protein-coding genes (CDS), 38 tRNA genes and 8 rRNA genes. By comparing the sequence differences between these two cp genomes, it was found that they are relatively conserved in terms of overall gene content and gene arrangement. However, a total of 684 SNPs and 2664 indels were still identified, with *ycf*1, *clp*P, and *trn*K-UUU protein-coding genes having the highest number of SNPs and indels. In further comparative analyses among the six cp genomes in Malaxidinae, significant sequence divergences were identified in the intergenic regions, namely *rps*16–*trn*Q-UUG, *trn*S-GCU–*trn*G-GCC, *rpo*B–*trn*C-GCA, *trn*E-UUC–*trn*T-GGU, *trn*F-GAA–*trn*V-UAC, *atp*B–*rbc*L, *pet*A–*psb*J, *psb*E–*pet*L, *psb*B–*psb*T, *trn*N-GUU–*rpl*32, *trn*V-GAC–*rps*7, and *rps*7–*trn*L-CAA, and five coding regions, including *mat*K, and *rpo*C2, *ycf*1, and two *ycf*2 genes. Phylogenetic analysis indicated that *Blepharoglossum* and *Oberonia* form a highly supported sister group relationship. Our results are consistent with previous studies and present increased resolution among major clades.

## 1. Introduction

The orchid genus *Blepharoglossum* (Schltr.) L. Li of Malaxidinae (Orchidaceae, Malaxideae), includes species previously ascribed to *Liparis* Rich. s.l. or *Cestichis* Thouars ex Pfitzer [1,2]. The most recent phylogenetic study on 50 species of Malaxideae presented a phylogeny of *Cestichs* and allied genera using ITS and *mat*K sequence data [3]. *Blepharoglossum* was placed as sister to *Oberonia* Lindl. with weak support. Based on morphological characters, they observed a clear difference with regard to their rachis characters. *Blepharoglossum* species have terete rachises; in contrast, species of *Cestichis* and *Stichorkis* Thouars possess slightly or strongly flattened rachises. In addition, *Blepharoglossum* species have a deeply bilobed lip apex with margins that are lobulated and ciliolate to deeply lacerate. Based on these findings, *Blepharoglossum* was elevated to full generic status. *Blepharoglossum* comprises approximately 39 species that are mostly distributed in tropical Asia–Pacific islands, including Peninsular Malaysia, Borneo, Sumatra, Java, Sulawesi, the Philippines, Papua New Guinea, New Caledonia, and Fiji, with some species found in Myanmar, Thailand, Vietnam, and Taiwan and Hainan Islands in China [3,4].

The *Blepharoglossum* is a precious orchid genus, whose species are all listed in Appendix I of the Convention on International Trade in Endangered Species Wild Fauna and Flora (CITES). Due to the destruction of its habitat and people’s enthusiasm for the mining of orchid plants, it is further lacking resources. In Flora of China, the *Blepharoglossum* encompasses 5 distinct species, namely *B. condylobulbon* (Rchb.f.) L. Li, *B. elegans* (Lindl.) L. Li, *B. fissilabrum* (T. Tang and F.T. Wang) L. Li, *B. grossum* (Rchb.f.) L. Li, and *B. latifolium* (Lindl.) L. Li. Despite the fact that they serve as an important resource for orchids, the cp genomes of these five species have not been previously characterized. Therefore, it is imperative to conduct a comprehensive analysis to explore the sequence variability within the cp genome of this genus, as it holds significant implications for conservation efforts. Moreover, there is still uncertainty regarding the intergeneric relationships within Malaxidinae at present. Although the *Blepharoglossum* was previously and for a long time placed as *Liparis* s.l. or *Cestichis* due to the resemblances in floral morphology, the molecular systematics and morphology evidence indicated that the *Blepharoglossum* clade and other *Liparis* alliances had distant relationships and significantly different characteristics. Thus, it had been separated from epiphytic *Liparis* s.l. and elevated to a generic status in recent years [3]. Nevertheless, the taxonomic status of *Blepharoglossum* and the phylogenetic relationships within its relevant groups have been controversial due to their limited resolution with traditional sequences from one or a few markers in previous studies that await stronger evidence to support them. This is probably because its relatives have a notoriously complicated taxonomy. Therefore, the development of more effective genetic resources on the basis of high-resolution molecular markers is necessary for further phylogenetic studies of *Blepharoglossum*.

The chloroplast (cp) genomes have been extensively employed in various areas of research, such as in species identification, evolutionary assessments, and in the development of molecular markers [5,6,7]. Research has demonstrated that cp genomes offer distinct advantages for resolving the phylogenetic relationships of angiosperms, primarily due to their high-copy nature, structural simplicity, and highly conserved gene content [8,9]. In Orchidaceae, for instance, some genera exhibit unresolved phylogenetic relationships and uncertain monophyletic status with traditional sequences from a single or few markers, such as nrDNA-ITS and *mat*K. However, the cp genome data have demonstrated remarkable advantages in resolving the developmental relationships and systematic positions of complex taxa, resulting in significant improvements in accuracy and reliability [10,11]. Moreover, a comparative analysis of cp genomes presents a promising avenue for investigating sequence variations and identifying mutation hotspots. The identification of such regions from cp genomes can serve as informative molecular markers for species delimitation and genetic analyses of populations [12,13].

In this study, we sequenced and assembled the complete cp genomes of two *Blepharoglossum* species, *B. elegans* and *B. grossum*, which have not been previously characterized. We then analyzed their differences and compared sequence features with those of four other species in allied genera within Malaxidinae. The aims of this investigation were two-fold: (1) to elucidate and contrast the cp genomes of *Blepharoglossum*, identifying variations among these six species, and (2) to reconstruct a robust phylogenetic tree of *Blepharoglossum* and its closely related taxa using cp genome sequences, thereby elucidating their phylogenetic relationships and verifying their taxonomic status within Malaxidinae. Our cp genome data presented here will provide an informative and valuable genomic resource for facilitating species authentication and evolutionary analysis in the Orchidaceae.

## 2. Materials and Methods

### 2.1. Sampling and DNA Extraction

The samples of *B. elegans* and *B. grossum* were obtained from the living collections of the greenhouse of South China Botanical Garden, Chinese Academy of Sciences (SCBG, CAS, 23°10.858′ N, 113°21.136′ E, 27 m), which were introduced from the field populations in Hainan, China in 2021. The voucher specimens had been lodged at the herbarium of South China Botanical Garden (IBSC), under collection numbers YWT042 (*B. grossum*) and YWT043 (*B. elegans*). Total genomic DNA was extracted from fresh leaf tissues using a Plant Genomic DNA kit (Tsingke Biological Technology, Beijing, China) following the manufacturer’s instructions.

### 2.2. DNA Library Preparation and Sequencing

The total DNA was fragmented and used to set up short-insert libraries and the qualified libraries were sequenced with PE150 bp on the DNBseq-2000RS platform at Beijing Genomics Institute (Wuhan, China). There was ~3.0 Gb total amount of data and about 20× sequencing depth. Then we used SOAPnuke software [14] to filter the low-quality data of the generated paired-end raw reads to obtain clean data, which could be used for subsequent analysis. Filtering standard: remove reads with an N base content exceeding 5%; remove reads with a low quality (mass value less than or equal to 5) base count of 50%; remove reads contaminated with adapter.

### 2.3. Chloroplast Genome Assembly and Annotations

Clean data were used for assembling the complete cp genomes by using GetOrganelle v1.6.2e [15]. We downloaded several cp genomes closest to *Blepharoglossum* from NCBI GenBank database as reference sequences for assembly, i.e., *Liparis auriculata* Blume ex Miq. (GenBank accession no.: MN200365), *Liparis bootanensis* Griff. (GenBank accession no.: MN627759), *Liparis nervosa* (Thunb.) Lindl. (GenBank accession no.: MN641753), and *Liparis pingtaoi* (G.D. Tang, X.Y. Zhuang and Z.J. Liu) J.M.H. Shaw (GenBank accession no.: MN627758). The assembled cp genomes were annotated with PGA [16], then manually adjusted and confirmed using Geneious R9.0.2 [17]. The circular cp genome maps were visualized by OrganellarGenomeDRAW (OGDRAW) version 1.3.1 [18]. Utilizing Geneious R9.0.2, we identified the number of single-nucleotide polymorphisms (SNPs) and insertions–deletions (Indels) that existed between the two *Blepharoglossum* species under analysis, using *B. grossum* as a reference sequence.

### 2.4. Genome Comparison and Phylogenetic Identification

In addition to the newly sequenced genomes of two *Blepharoglossum* species noted above, we downloaded four published sequences of their allies from the NCBI database, i.e., *L. bootanensis* (GenBank accession no.: MN627759), *L. viridiflora* (Blume) Lindl. (GenBank accession no.: MW691170), *Stichorkis gibbosa* (Finet) J.J. Wood (GenBank accession no.: OM759993), and *Oberonia seidenfadenii* (H.J. Su) Ormerod (GenBank accession no.: MN414241) in order to identify the genetic and genomic features among different genomes. The location of genes on the IR/SC boundaries were visualized by IRscope [19]. The divergent regions were accessed using mVISTA online program in Shuffle-LAGAN mode with the cp genome sequence of *O. seidenfadenii* selected as a reference [20].

To gain a more thorough understanding of the intergeneric relationships between *Blepharoglossum* and related genera, we reconstructed the phylogenetic tree of Malaxidinae based on 25 cp genome data representing 24 species. In addition to the 2 newly sequenced *Blepharoglossum* cp genomes, another 23 cp genomes were obtained from the NCBI GenBank database. Sequences of three species, i.e., *Epipactis purpurata* Sm., *Neottia japonica* (Blume) Szlach., and *Cephalanthera humilis* X.H. Jin, were selected as outgroups based on earlier phylogenetic studies [21]. All the 25 genome sequences were aligned and adjusted using the plug-in MAFFT Alignment in Geneious R9.0.2. We constructed the phylogenetic tree using IQ-TREE [22]. The best-fit model was GTR + F + I + G4 in this analysis and we validated it by 1000 SH-aLRT tests and ultrafast tests bootstrap approach (UFboot) [23,24]. The generated phylogenetic tree was visualized through the utilization of Figtree software. Each branch of the phylogenetic tree contains SH-aLRT and UFboot support rates. If SH-aLRT ≥ 80% and UFboot ≥ 95%, it is considered to have received good support and the branch results are reliable.

## 3. Results and Discussion

### 3.1. Chloroplast Genome Features

The chloroplast genome sequences of *B. elegans* and *B. grossum* that were obtained in this study have been deposited in the NCBI GenBank database under the accession numbers OP859143 and OP859144, respectively. Both *Blepharoglossum* cp genomes exhibited the typical circular and quadripartite structure characteristic of most angiosperms (Figure 1), which consists of a pair of inverted repeat regions (IRa and IRb) that are separated by a large single copy (LSC) and a small single copy (SSC). The cp genome sequences of *Blepharoglossum* are 156,813 bp (*B. elegans*) and 157,549 bp (*B. grossum*) in size. The cp genome of *B. elegans* was extremely similar to that of *B. grossum*. The length of the LSC regions are 84,984 bp (*B. elegans*) and 85,588 bp (*B. grossum*), the SSC regions are 17,731 bp (*B. elegans*) and 17,765 bp (*B. grossum*), and the IR regions are 27,049 bp (*B. elegans*) and 27,098 bp (*B. grossum*), respectively (Table 1). The complete set of 133 genes, comprising 87 protein-coding sequences (CDS), 38 transfer RNA genes (tRNAs), and 8 ribosomal RNA genes (rRNAs), were encoded by each of the genomes (Table 2). Except for *O. seidenfadenii*, the length and gene content of the cp genomes of the other five species varied little and were very close. Among these cp genomes of 6 species, the longest was 158,325 bp in *L. bootanensis* and the shortest was 143,062 bp in *O. seidenfadenii*. The main variations among them are the length of the SSC and IR regions. Corresponding to the smaller size, the cp genome of *O. seidenfadenii* was annotated with only 120 genes, including 74 protein-coding genes, 38 tRNA, and 8 rRNA (Table 1).

### 3.2. SNPs and Indels Detection between B. grossum and B. elegans

To further explore the genetic disparities of the two *Blepharoglossum* plastomes, SNPs (single nucleotide polymorphisms) and indels (insertions/deletions) were identified with the cp genome of *B. grossum* as a reference. Compared to the *B. grossum* reference genome, 684 SNPs and 2664 indels were detected in *B. elegans* (Appendix A). In total, there are 321 SNPs in 63 protein-coding genes (CDS) and 364 SNPs in intergenic regions (IGS) detected between the cp genomes of *B. elegans* and *B. grossum* (Table 3). In terms of protein-coding genes, the number of SNPS is maximal in the *ycf*1, *ndh*A, *clp*P, and *trn*K-UUU genes, with 43, 20, 24, and 23, respectively (Figure 2A). A total of 964 insertions and 1700 deletions were detected between the two cp genomes of *Blepharoglossum* species (Appendix A). The results showed that 2162 indels (752 insertions and 1210 deletions) were located in intergenic spacer regions, while the other 502 indels (212 insertions and 290 deletions) were distributed in 18 protein-coding genes, including *ycf*2, *trn*A-UGC, *ycf*1, *ndh*A, *rpl*16, *pet*D, *pet*B, *clp*P, *rpl*33, *trn*V-UAC, *trn*L-UAA, *trn*S-GGA, *rpo*C1, *rpo*C2, *atp*F, *trn*G-GCC, *rps*16, and *trn*K-UUU (Figure 2B). Among them, indels were mostly found in the *ycf*2, *ycf*1, *clp*P, *atp*F, *rps*16, and *trn*K-UUU genes, corresponding to 222, 21, 26, 33, 79, and 28 respectively. The distributions of SNPs and indels is shown in Figure 2C,D. The number of SNPs located in the LSC, SSC and IRs regions accounted for 69.2%, 18.1%, and 12.7%, respectively. The number of indels in the LSC, IRs and SSC regions accounted for 82.5%, 12.4% and 5.1%, respectively.

Overall, the distribution patterns of SNPs and indels in the two cp genomes are similar, with a large number of variations present in the LSC region and relatively few polymorphic sites in the SSC and IRs regions. Furthermore, the majority of mutations in these genomes existed in the non-coding regions, mainly concentrated in the intergenic spacer regions, and some coding regions also contained a considerable number of mutation sites. These SNPs and the indels identified herein provide a reliable estimate of genome-wide genetic variations of the two *Blepharoglossum* species.

### 3.3. Details of the IR Contraction and Expansion

The IR/SC boundary regions and adjacent genes were compared among the chloroplast genomes of *Blepharoglossum* and its allies (*L. bootanensis*, *L. viridiflora*, *S. gibbosa*, and *O. seidenfadenii*). In all these cp genomes, the *rpl*22 gene crossed the LSC/IRb border. The IRa/LSC junction was located between *rps*19 and *psb*A, and variation between *psb*A and the Ira/LSC border ranged from 85 bp to 185 bp (Figure 3). However, the IR/SSC boundaries in these genomes exhibited obvious differences. The *ndh*F gene was found to be absent at the IRb/SSC boundary in *O. seidenfadenii*. In particular, with the exception of *O. seidenfadenii*, the *ycf*1 gene crossed the SSC/IRa junction in all compared species and extended 5375 bp–5534 bp into the IRa regions, creating a pseudogene (ψ*ycf*1), with the fragments varying from 1055 bp to 1159 bp in the IRb region. In contrast, the *ycf*1 gene of *O. seidenfadenii* was nearly completely situated in the SSC region, extending only 2 bp to the IR region, and thus, no pseudogene was created at the border. Furthermore, overlaps between the ψ*ycf*1 fragment and the *ndh*F gene were detected at the IRb/SSC junction. In *B. elegans*, *B. grossum*, and *S. gibbosa*, the *ndh*F gene was located 65 bp–68 bp away from IRb/SSC, whereas in *L. bootanensis* and *L. viridiflora*, it was only 9 bp away, which also had a 9 bp overlap with ψ*ycf*1.

Previous studies have revealed that the *ndh*F gene has been pseudogenized or lost in most orchid cp genomes, which is correlated with shifts in the position of the junction of the IR/SC regions [25,26,27]. The expansion and contraction of IR boundary regions or gene loss could account for size variations in cp genomes [27,28]. Compared to the other five cp genomes, the cp genomes size in *O. seidenfadenii* was considerably shorter. The IR regions of *O. seidenfadenii* was only 24,278 bp in length, while the IR regions of the other species examined were much longer, approximately 27,000 bp in length (Table 2). The *ndh*F gene in the *O. seidenfadenii* cp genome is apparently lost. Gene loss could contribute to the adaptive evolution in response to a changing environment [29].

### 3.4. Comparative Analysis of the Chloroplast Genomes

The sequences of the two *Blepharoglossum* cp genomes, together with four other published cp genomes of related species, were aligned and compared using mVISTA with the annotated *O. seidenfadenii* cp genome (GenBank accession no.: MN414241) as a reference. Our analyses showed that higher divergences were distributed in the non-coding regions compared to the protein-coding regions. Expectedly, significant divergences were identified within non-coding intergenic spacer (IGS) regions. Additionally, for both coding and non-coding regions, the LSC and SSC regions exhibited higher sequence variations than the two IR regions. Notably, we found two separate deletions of about 240 bp in the *psb*E–*pet*L region and 473 bp in the *trn*E-UUC–*trn*T-GGU region for both of *B. elegans* and *B. grossum* cp genomes. In addition, *B. elegans* carried a unique deletion of 400 bp in the *ycf*4–*cem*A intergenic region. Most of the genetic variabilities were identified in the intergenic spacers, such as *rps*16–*trn*Q-UUG, *trn*S-GCU–*trn*G-GCC, *rpo*B–*trn*C-GCA, *trn*E-UUC–*trn*T-GGU, *trn*F-GAA–*trn*V-UAC, *atp*B–*rbc*L, *pet*A–*psb*J, *psb*E–*pet*L, *psb*B–*psb*T, *trn*N-GUU–*rpl*32, *trn*V-GAC–*rps*7, and *rps*7–*trn*L-CAA. Somewhat higher sequence variations in the coding regions were observed, including *rpo*C2, *ycf*1, and two *ycf*2 genes (Figure 4). The significant divergences detected in both the intergenic spacer regions and the genes align with similar findings from other cp genomes of orchid species [27,30], which could be used as potential molecular markers.

### 3.5. Phylogenetic Analysis

Overall, the topology of the phylogenetic tree inferred from our maximum likelihood (ML) analyses was largely consistent with those from previous studies using molecular data from a few markers of nuclear and plastid regions [3,31,32]; however, the full cp genome sequences produced a more resolved and better-supported phylogeny (Figure 5). Similarly, the members of Malaxidinae in this study were clearly clustered into two distinct lineages corresponding to the epiphytes and terrestrials. The two representatives of *Blepharoglossum* formed a well-supported monophyletic lineage, sister to *Oberonia* with high support. Our results revealed that *Blepharoglossum* and *Oberonia* were more closely related to each other than to any other groups traditionally recognized within the Malaxidinae. The epiphytic *Liparis* alliance was reconfirmed to form a polyphyletic assemblage of three distant lineages (the core *Cestichis*, *Stichorkis*, and *Blepharoglossum*) that have evolved independently.

## 4. Conclusions

This study first shed light on the structure and content of the complete chloroplast genomes of two *Blepharoglossum* species. A total of 4 genes (*ycf*1, *ndh*A, *clp*P, and *trn*K-UUU) exhibited more than 20 SNPs, while 6 genes (*ycf*2, *ycf*1, *clp*P, *atp*F, *rps*16, and *trn*K-UUU) had indel numbers exceeding 20, which suggests that these genes could serve as promising molecular markers for studying *Blepharoglossum*, aiding in practical applications such as resource conservation and species identification. We also offered information on the sequence divergences in these cp genomes and the four other published Malaxidinae cp genomes. Among these 6 cp genomes, genomic divergences were detected as promising DNA barcodes, including 12 intergenic regions and 5 protein-coding genes. In particular, two separate deletions occurred in the two *Blepharoglossum* cp genomes which were not found in the other allies. The positions of the IR/SSC boundaries associated with the *ycf*1 genes and the *ndh*F genes were quite variable for these six cp genomes. Additionally, the phylogeny inferred from the complete cp genomes greatly improved the phylogenetic resolution of the *Stichorkis* and *Blepharoglossum* clades and will likely contribute to our knowledge of the generic circumscription within Malaxidinae.

## Figures and Tables

**Figure 1 genes-14-01069-f001:**
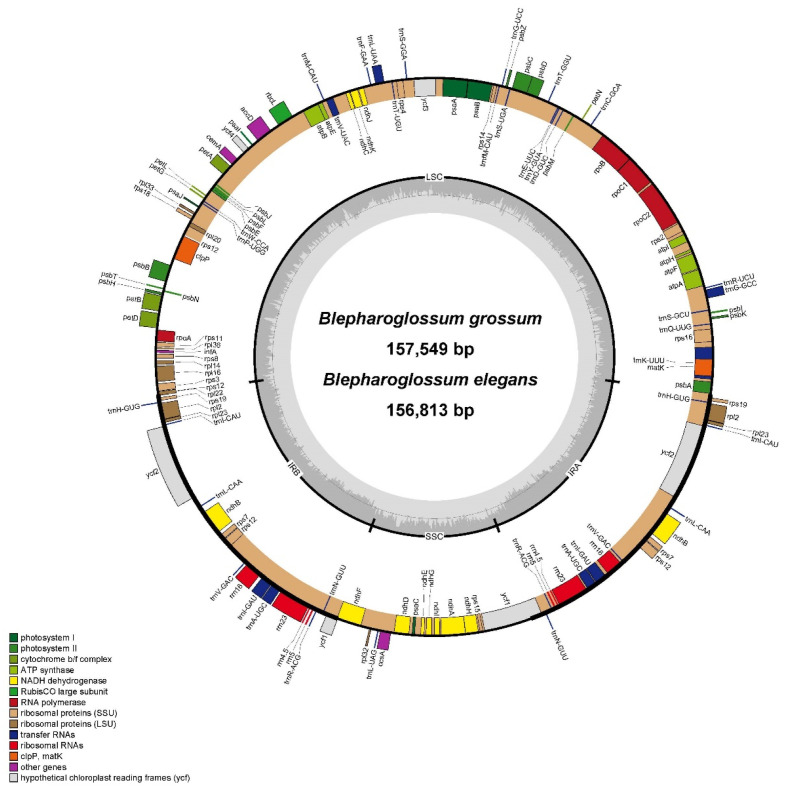
Gene map of chloroplast genomes from *B. elegans* and *B. grossum*. The genes transcribed in the clockwise direction are depicted inside the circle, while those transcribed in the counterclockwise direction are shown outside. Functional groups of genes are distinguished by different colors. The darker gray color in the inner circle represents the GC content, while the lighter gray color corresponds to the AT content.

**Figure 2 genes-14-01069-f002:**
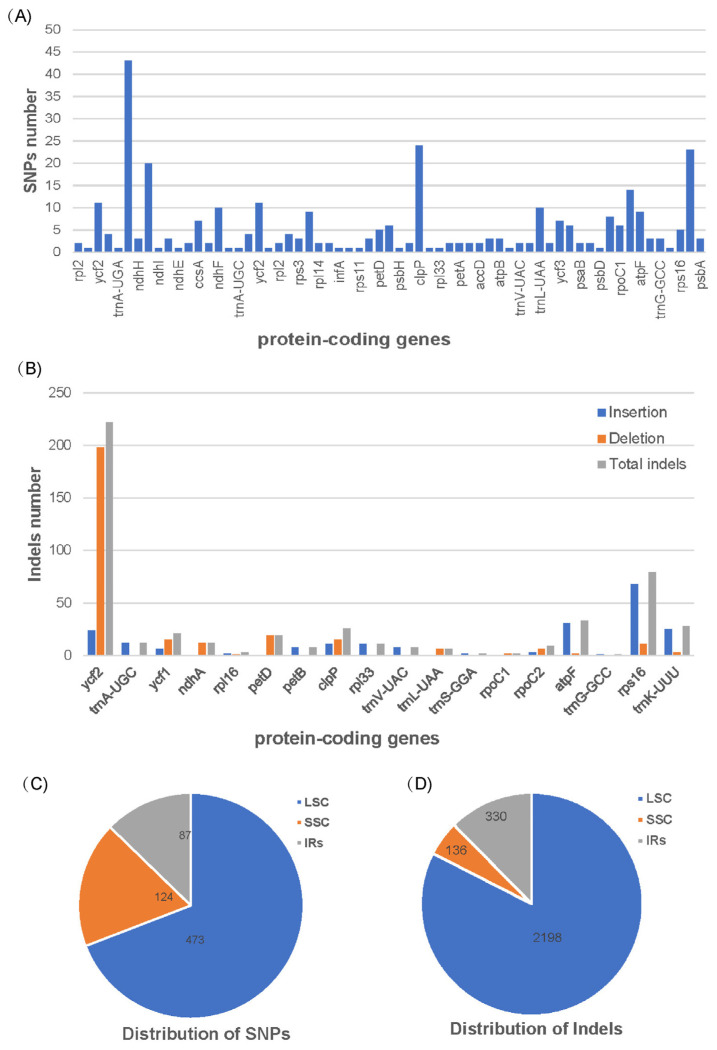
SNPs and indels statistics between *B. grossum* and *B. elegans*. (**A**) SNPs number in various protein-coding genes. (**B**) Insertion, deletion, and total indel statistics in various protein-coding genes. (**C**) Location of SNPs. (**D**) Location of Indels.

**Figure 3 genes-14-01069-f003:**
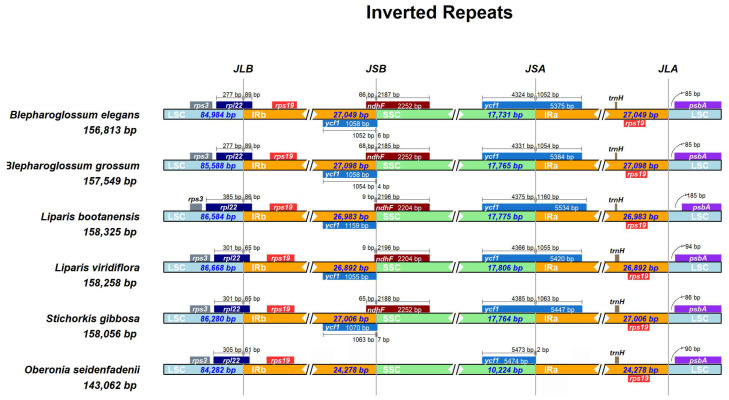
Comparison of border positions of LSC, SSC, and IRs regions among the six cp genomes. The diagram depicts genes as boxes, with the distance between the genes and the boundaries indicated by the number of bases, unless the gene coincides with the boundary. Gene extensions are displayed above the boxes. JLB, JSB, JSA, and JLA correspond to boundaries of LSC/IRb, SSC/IRb, SSC/IRa, and LSC/IRa, respectively.

**Figure 4 genes-14-01069-f004:**
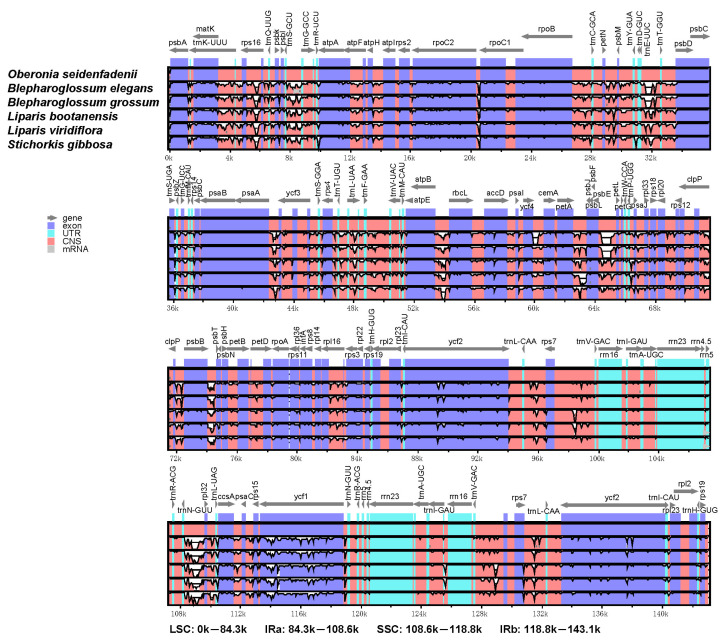
Global alignment of chloroplast genomes of the six species by mVISTA using *O. seidenfadenii* as the reference. Genome regions are color coded. Blue and red areas indicate untranslated regions (UTR) and the conserved non-coding sequences (CNS) regions, respectively.

**Figure 5 genes-14-01069-f005:**
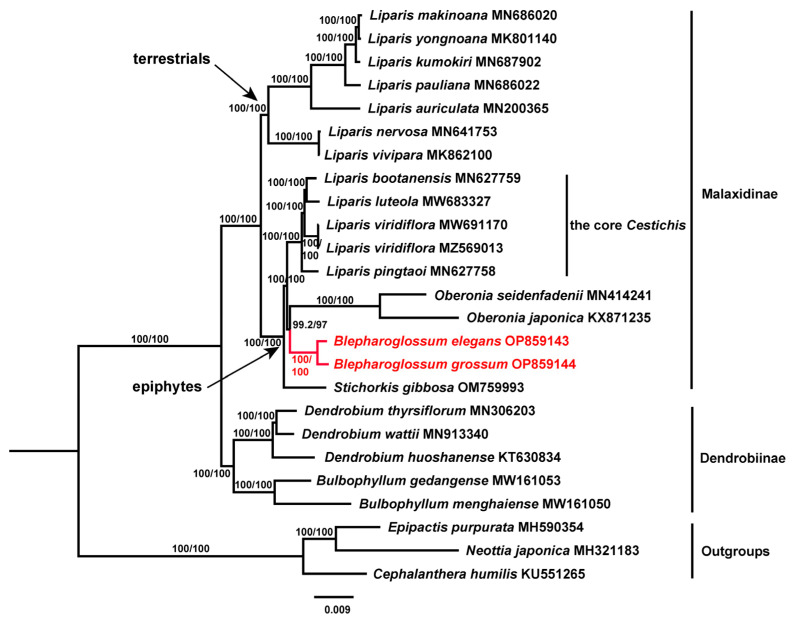
The maximum likelihood (ML) phylogenetic tree constructed based on complete cp genome sequences, showing the phylogenetic relationships of *Blepharoglossum* and its related genera in Malaxidinae. Red color indicates taxa of *Blepharoglossum*. Branch support values are shown in order SH-aLRT/UFboot.

**Table 1 genes-14-01069-t001:** Summary of the chloroplast genome characteristics of six Malaxidinae species.

Species		*B. elegans*	*B. grossum*	*L. bootanensis*	*L. viridiflora*	*S. gibbosa*	*O. seidenfadenii*
GenBank ID	data	OP859143	OP859144	MN627759	MW691170	OM759993	MN414241
Length (bp)	Total	156,813	157,549	158,325	158,258	158,056	143,062
	LSC	84,984	85,588	86,584	86,668	86,280	84,282
	SSC	17,731	17,765	17,775	17,806	17,764	10,224
	IR	27,049	27,098	26,983	26,892	27,006	24,278
Gene number	Total	133	133	133	133	133	120
	CDS	87	87	83	83	86	74
	tRNA	38	38	38	38	38	38
	rRNA	8	8	8	8	8	8
GC content (%)	Total	36.8	36.8	36.9	36.8	36.9	37.1
	LSC	34.3	34.4	34.5	34.3	34.4	34.4
	SSC	29.6	29.8	30.0	29.8	29.7	27.9
	IR	43.0	43.1	43.1	43.1	43.1	43.7

**Table 2 genes-14-01069-t002:** Gene groups with different functions in the chloroplast genomes of *B. elegans* and *B. grossum*.

Gene Group	Gene Name
photosystem I	*psa*A, *psa*B, *psa*C, *psa*I, *psa*J
photosystem II	*psb*A, *psb*B, *psb*C, *psb*D, *psb*E, *psb*F, *psb*H, *psb*I, *psb*J, *psb*K, *psb*L, *psb*M, *psb*N, *psb*T, *psb*Z
NADH dehydrogenase	*ndh*A*, *ndh*B*(2), *ndh*C, *ndh*D, *ndh*E, *ndh*F, *ndh*G, *ndh*H, *ndh*I, *ndh*J, *ndh*K
Cytochrome b/f	*pet*A, *pet*B*, *pet*D*, *pet*G, *pet*L, *pet*N
ATP synthase	*atp*A, *atp*B, *atp*E, *atp*F*, *atp*H, *atp*I
rubisco	*rbc*L
Proteins of large ribosomal subunit	*rpl*14, *rp*l16*, *rpl*2*(2), *rp*l20, *rpl*22, *rpl*23(2), *rpl*32, *rpl*33, *rpl*36
Proteins of small ribosomal subunit	*rps*11, *rps1*2**(2), *rps*14, *rps*15, *rp*s16*, *rps*18, *rps*19(2), *rps*2, *rps*3, *rps*4, *rps*7(2), *rps*8
RNA polymerase	*rpo*A, *rpo*B, *rpo*C1*, *rpo*C2
Ribosomal RNAs	*rrn*16(2), *rrn*23(2), *rrn*4.5(2), *rrn*5(2)
Transfer RNAs	*trn*A-UGC*(2), *trn*C-GCA, *trn*D-GUC, *trn*E-UUC, *trn*F-GAA, *trn*G-GCC*, *trn*G-UCC*, *trn*H-GUG(2), *trn*I-CAU(2), *trn*I-GAU*(2), *trn*K-UUU*, *trn*L-CAA(2), *trn*L-UAA*, *trn*L-UAG, *trn*M-CAU, *trn*N-GUU(2), *trn*P-UGG, *trn*Q-UUG, *trn*R-ACG(2), *trn*R-UCU, *trn*S-GCU, *trn*S-GGA, *trn*S-UGA, *trn*T-GGU, *trn*T-UGU, *trn*V-GAC(2), *trn*V-UAC*, *trn*W-CCA, *trn*Y-GUA, *trnf*M-CAU
Maturase	*mat*K
Protease	*clp*P**
Envelope membrane protein	*cem*A
Acetyl-CoA carboxylase	*acc*D
c-type cytochrome synthesis gene	*ccs*A
Translation initiation factor	*inf*A
Conserved hypothetical chloroplast ORF	*yc*f1(2), *ycf*2(2), *ycf*3**, *ycf*4

Note: Gene*: Gene with one intron; gene**: gene with two introns; gene(2): number of copies of multi-copy genes.

**Table 3 genes-14-01069-t003:** Distribution of SNPs and indels in different genic regions.

	SNPs	Indels	
		Insertions	Deletions
CDS	321	212	290
IGS	364	752	1410
Total	684	964	1700

## Data Availability

The genome sequence data of *B. elegans* and *B. grossum* of this study are openly available in NCBI GenBank database (https://www.ncbi.nlm.nih.gov/, accessed on 1 March 2023) under accession numbers of OP859143 and OP859144, respectively.

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
