# Peer review of "The Complete Chloroplast Genomes of Blepharoglossum elegans and B. grossum and Comparative Analysis with Related Species (Orchidaceae, Malaxideae)"

_genes, 2023, doi:10.3390/genes14051069_

Round 1

Reviewer 1 Report

Recommendation: Major revision

Dear Authors,

It was a pleasure to read your manuscript. “The complete chloroplast genomes of Blepharoglossum elegans and B. grossum and comparative analysis with related species (Orchidaceae, Malaxideae)”.

I feel it should be accepted for the publication after undergoing some major revisions.

As you state, the data presented in this study is an important to the genomic resource for facilitating the species authentication and evolutionary analysis in the family Orchidaceae family. I am curious about the herbarium in which you deposited the voucher material and also I could not find its official herbarium code in the Index Herbariorum. There is a lack of why this two species were chosen for the decode their chloroplast genome and what is the importance and applications of these two species?.

There are some areas for major revision:

  1. I still believe it is important for to add details about the quality of the Illumina Sequencing reads. Raw data from Illumina reads, about trimming or quality control program, quantity of data obtained and how much data used for the assembly, and its coverage.
  2. Before mentioning the plant species name, we should always follow the currently accepted name.
  3. This is a full length research paper, and there should be discuss your results briefly in detail with supportive previous literature. Instead I am feeling like short communication or genome announcement article. For assembly the authors choosen Liparis species in GetOrganelle but other than that no information species marker among Liparis species and there is lack of discussion part.

Query 1:

Plant species collection information.

I observed that the authors only mentioned distribution details.

Include the details of place of collections with GPS co-ordinates,

Query 2:

Voucher Specimen Information.

I am curious about the herbarium in which you deposited the voucher material, because I could not find any information about voucher specimen/herbarium deposition.

It is advisable to submit the voucher specimen to a Indexed Herbariorum repository.

Query 3:

Currently accepted species name

The species Blepharoglossum elegans and Blepharoglossum grossum, both are synonyms name. Authors are request to check the currently accepted name. Example: https://powo.science.kew.org/

Query 4:

Illumina sequence data QC and assembly details

What is the trimming /quality control program did you use? what total amount of data received from the Illumina reads ? What was the average length of pair end read? what was the mean, median, and range for the depth of coverage of the chloroplast genome?

Query 6:

Chloroplast genome comparision

I observed there is a size variation in LSC, SSC, IRS between two species. But the no. of gene content is same. Can author address this in which region contains variation in nucleotide and mentioned the variants type (SNP or InDels) in details or as table ?

Query 7:

There is a lack of information about plant species choosen by the authors for this study.

Add more details about plant species, why particularly these two species choosen among other species. is they contain any medicinally important compounds or applications ?

Reviewer 2 Report

 The complete chloroplast genomes of Blepharoglossum elegans and B. grossum and comparative analysis with related species (Orchidaceae, Malaxideae)

 Wenting Yang, Kunlin Wu, Lin Fang, Songjun Zeng and Lin Li

 Blepharoglossum is a rare orchid genus mainly found on tropical islands in the Pacific.

The authors sequenced the chloroplast genomes of Blepharoglossum elegans and B. grossum.  They then compared these 2 sequences with each other, and with 4 previously-sequenced plastomes of the Malaxidinae subtribe of orchids as well as those of 18 other plant species. They provide a detailed comparative analysis of structures of these genomes, and a phylogenetic analysis showing that Blepharoglossum and Oberonia form a highly-supported sister group whereas previous studies only provided weak support for this relationship.

My only concern is that I would like to see the authors explain better why this study was worth performing. Do these studies provide better insight into how plants evolve?  Do these plants have any economic or medical importance?

Otherwise, it seems to be a solid report. I therefore recommend publication after addressing the major question and correcting some minor issues noted below.

Line 30: please change “including” to “includes.”

Line 52:  please change “compared with those of other four species of allied genera” to “compared with those of four other species of allied genera.”

Line 59:  please change “Chinese of Academy of Sciences” to “Chinese Academy of Sciences.”

Line 77: please change “manual” to “manually.”

Lines 97-102 are hard to understand and should be rewritten.

Line 105: did you sequence the entire genomes of these 2 species? If not, please change “genomes” to “plastomes.”

Reviewer 3 Report

The manuscript describes on the characterization, genome comparative and phylogenomics of Blepharoglossum elegans and B. grossum. Proofreading is required.

L16 "unique" means that you do not include the duplicated genes in the IR region. I do not think it is appropriate in this context.

L43-45 What controversies are they? What previous studies? Weak problem statement.

L45 This should be a new paragraph after the above comments are being included. However, the description on cp genome is too short. The authors should describe successful stories using cp genome sequences in Orchidaceae.

L53 The authors started with "chloroplast (cp)" genomes, then here, "plastome". please uniform them. Choose one. Check throughout.

L55 -aceae indicates it as family. remove the word "family" to avoid redundancy. Same goes to genus and subtribe.

L66 DNBSeq, which model?

L67 3 Gb is not the depth. depth should be in times. Gb is size of raw NGS data.

L67-70 grammar

L72 version of pipelines/softwares. Check throughout.

L74 Please indicate "GenBank accession no.: xxxxx" for all the records.

L78 visuallized

L87 with the cp genome sequence of O. seidenfadenii selected as reference

L90 reconstructed the phylogenetic tree of. Avoid using the word phylogeny. You cannot reconstruct phylogeny; it is there since the beginning.

L96-102 grammar

L112-115 There should not be a range for these findings as there are only two species (two individuals). The authors should just state them accordingly, instead of giving a range.

L116 unique?

L116 instead of PCG, please use CDS.

L117-122 grammar

L130 Details of the

References - please check format throughout. to name a few: [21] has all the first letter in capital, while others do not; [17] AM J. Bot.; [9], etc

L252 The reference of GetOrganelle is incorrect. Please carefully check ALL of them.

Round 2

Reviewer 1 Report

Dear Authors,

It was a pleasure to read the revised manuscript entitled “The complete chloroplast genomes of Blepharoglossum elegans and B. grossum and comparative analysis with related species (Orchidaceae, Malaxideae)”.

Thanks to all authors for considering my suggestions and accepting my comments on your manuscript. I hope the authors are clearly gone through all the comments and changes made to correct it on the revised manuscript. I am satisfied with the author's responses.

 Thank you